

# Interrogating COVID-19 vaccine intent in the Philippines with a nationwide open-access online survey

Alexandria Caple[1], Arnie Dimaano[2], Marc Martin Sagolili[2], April Anne Uy[2], Panjee Mariel Aguirre[3], Dean Lotus Alano[2], Giselle Sophia Camaya[2], Brent John Ciriaco[3], Princess Jerah Mae Clavo[2], Dominic Cuyugan[2], Cleinne Florence Geeseler Fermo[2], Paul Jeremy Lanete[2], Ardwayne Jurel La Torre[3], Thomas Loteyro[3], Raisa Mikaela Lua[2], Nicole Gayle Manansala[2], Raphael Willard Mosquito[2], Alexa Octaviano[2], Alexandra Erika Orfanel[2], Gheyanna Merly Pascual[2], Aubrey Joy Sale[2], Sophia Lorraine Tendenilla[2], Maria Sofia Lauren Trinidad[2], Nicole Jan Trinidad[2], Daphne Louise Verano[2] and Nicanor Austriaco[2,4]

[1] Psychology, Providence College, Providence, RI, United States of America
[2] Biological Sciences, University of Santo Tomas, Manila, Philippines
[3] Advertising Arts, University of Santo Tomas, Manila, Philippines
[4] Biology, Providence College, Providence, RI, United States of America

## ABSTRACT

To mitigate the unprecedented health, social, and economic damage of COVID-19, the Philippines is undertaking a nationwide vaccination program to mitigate the effects of the global pandemic. In this study, we interrogated COVID-19 vaccine intent in the country by deploying a nationwide open-access online survey, two months before the rollout of the national vaccination program. The Health Belief Model (HBM) posits that people are likely to adopt disease prevention behaviors and to accept medical interventions like vaccines if there is sufficient motivation and cues to action. A majority of our 7,193 respondents (62.5%) indicated that they were willing to be vaccinated against COVID-19. Moreover, multivariable analysis revealed that HBM constructs were associated with vaccination intention in the Philippines. Perceptions of high susceptibility, high severity, and significant benefits were all good predictors for vaccination intent. We also found that external cues to action were important. Large majorities of our respondents would only receive the COVID-19 vaccines after many others had received it (72.8%) or after politicians had received it (68.2%). Finally, our study revealed that most (21%) were willing to pay an amount of PHP 1,000 (USD20) for the COVID-19 vaccines with an average willing-to-pay amount of PHP1,892 (USD38).

Corresponding author
Nicanor Austriaco,
naustria@providence.edu

## INTRODUCTION

On January 30, 2020, the Department of Health (DOH) of the Philippines reported its first case of COVID-19, a novel respiratory disease first identified in Wuhan, China, that is caused by the coronavirus, SARS-CoV-2 (*Xie et al., 2020*; *Zhu et al., 2020*; *Guan et al., 2020*). With widespread human-to-human transmission, the virus is highly contagious, and the COVID-19 pandemic is now of global concern (*Burki, 2020*; *Paules, Marston & Fauci, 2020*; *Case et al., 2021*; *Novelli et al., 2021*). As of August 25, 2021, there have been 1,883,088 confirmed cases and 32,492 deaths from COVID-19 reported by the DOH throughout the archipelago (https://doh.gov.ph/covid19tracker).

Vaccination has long been regarded as the most effective means for combating infectious disease (*Rappuoli et al., 2014*; *Sathyanarayana et al., 2020*). The Philippines began its national vaccine drive against COVID-19 on March 1, 2021, with the goal of vaccinating seventy million of its citizens by the end of the calendar year (*Inter-Agency Task Force for the Management of Emerging Infectious Disease, 2021*). One of the ongoing challenges for this campaign is the vaccine hesitancy among the Filipino people (*Alfonso et al., 2021*). Though immunization rates had been relatively high in the Philippines for many decades, the controversial 2016 rollout of the dengue vaccine, Dengvaxia, triggered significant drops in the rates of immunization as Filipino parents refused to have their children routinely vaccinated against polio, chicken pox, and tetanus (*Fatima & Syed, 2018*; *Smith, 2018*).

The Health Belief Model (HBM) posits that people are likely to adopt disease prevention behaviors and to accept medical interventions like vaccines if there is sufficient motivation and cues to action (*Rosenstock, Strecher & Becker, 1988*). Motivational factors include perceived susceptibility to and severity of the disease and perceived benefits of the vaccine. Cues to action include information, people, and events that nudge the individual towards vaccination. The HBM has been adopted as a conceptual framework that has been used to evaluate the beliefs and attitudes toward a diversity of vaccines including the influenza, human papillomavirus, and hepatitis B vaccines (*Teitler-Regev, Shahrabani & Benzion, 2011*; *Donadiki et al., 2014*; *Hu et al., 2017*; *Chen et al., 2019*). Moreover, several studies have shown that the HBM constructs can serve as an important predictor of influenza vaccination uptake (*Brewer et al., 2007*; *Shahrabani, Benzion & Yom Din, 2009*; *Shahrabani & Benzion, 2010*; *Tsutsui, Benzion & Shahrabani, 2012*). During the COVID-19 pandemic, the HBM was used to assess the root causes of COVID-19 vaccine hesitancy in the Asia-Pacific region and beyond (*Wong et al., 2020*; *Wong et al., 2021*; *Lin et al., 2020*; *Yu et al., 2021*; *Kabir et al., 2021*; *Shmueli, 2021*; *Huynh et al., 2021*; *Tao, Wang & Liu, 2021*; *Mahmud et al., 2021*; *Banik et al., 2021*).

In this study, we interrogated COVID-19 vaccine intent in the Philippines by deploying a nationwide open-access online survey, two months before the rollout of the national vaccination program. Based on the HBM framework, we hypothesized that acceptance of a COVID-19 vaccine depends upon beliefs about susceptibility to and severity of COVID-19, and beliefs about the perceived benefits of the vaccine. We also wanted to assess possible cues to vaccination for our Filipino respondents.

A majority of our 7,193 respondents (62.5%) indicated that they were willing to be vaccinated against COVID-19. Moreover, multivariable analysis revealed that HBM constructs were associated with vaccination intention in the Philippines. Perceptions of high susceptibility, high severity, and significant benefits were all good predictors for vaccination intent. We also found that external cues to action were important. Large majorities of our respondents would only receive the COVID-19 vaccines after many others had received it (72.8%) or after politicians had received it (68.2%). Finally, our study revealed that most (21%) were willing to pay an amount of PHP1,000 (USD20) for the COVID-19 vaccines with an average willing-to-pay amount of PHP1,892 (USD38). Based on these findings, we inaugurated the UST-CoVAX public awareness campaign that seeks to increase vaccine confidence in the Philippines by addressing the specific fears and concerns of our Filipino respondents and by sharing the personal vaccination testimonies of Filipinos around the world.

## METHODS

### Participants and survey design

The current study design was a cross-sectional, anonymous, web-based survey—developed using Qualtrics—conducted from January 15, 2021 to January 29, 2021. Our research team deployed an anonymous link *via* the social platforms of the University of Santo Tomas (UST) like Twitter and Facebook and university mailing lists including the UST School of Science and UST Student Council to distribute the survey. Participants were encouraged to distribute the survey link to their contacts throughout the country. The questionnaire was written in both Filipino and English. Responses used for data collection were limited to respondents who were at least 18 years old.

### Survey instrument

The survey consisted of questions and statements that assessed the following: (1) demographics, health status, and COVID-19 experience, (2) intent to receive a COVID-19 vaccine; (3) perceived susceptibility to and severity of COVID-19; (4) perceived benefits of a COVID-19 vaccine; (5) willingness to pay (WTP) for a COVID-19 vaccine; and (6) confidence in COVID-19 vaccines made in other countries.

Demographics, health status, and COVID-19 experience: Demographic information including age, gender, marital status, education, occupation, monthly income, and urban or rural location of residence were collected. Participants were also asked if they have an existing chronic condition, if they ever tested positive for COVID-19, and to indicate if they know someone who has tested positive for COVID-19.

Intent to receive a COVID-19 vaccine: Intention to receive a COVID-19 vaccine was assessed using a one-item question ("If a vaccine for COVID-19 is available in the Philippines, would you use it?") on a five-point scale ranging from 1 = 'definitely no' to 5 = 'definitely yes'. Responses were additionally recoded into two distinct categories: vaccine hesitant (responses included: 'definitely no', 'probably no', and 'unsure') and *not* vaccine hesitant (responses included: 'probably yes' and 'definitely yes').

Perceived susceptibility to and severity of COVID-19: HBM-derived items were used to assess individual beliefs about a COVID-19 vaccine. Questions posed to participants assessed perceived susceptibility of COVID-19 (two items), perceived severity of COVID-19 (three items), and cues to action (two items). All response items were on a four-point scale ranging from 'strongly agree' to 'strongly disagree'. For analysis purposes, all responses were coded as either 'agree' (responses included: 'strongly agree' and 'agree') or 'disagree' (responses included: 'strongly disagree' and 'disagree').

Perceived benefits of a COVID-19 vaccine: Perceived benefits were queried using two items. All response items were rated on a four-point scale ranging from 'strongly agree' to 'strongly disagree'. Similar to perceived susceptibility to and severity of COVID-19, all responses were coded as either 'agree' or 'disagree'. In addition to perceived benefits of a COVID-19 vaccine, respondents were also asked to rate–on a four-point scale ranging from 'strongly agree' to 'strongly disagree'–perceived barriers surrounding a COVID-19 vaccine (*e.g.*, 'I worry about the possible side-effects of the COVID-19 vaccine.'; 'I worry about fake COVID-19 vaccines.'). For analysis purposes, all responses were coded as either 'agree' or 'disagree'.

Willingness to pay for COVID-19 vaccine: Willingness to pay (WTP) was measured using a one-item question ("What is the maximum amount you are willing to pay for two doses of the COVID-19 vaccine?") on an eight-point scale ('PHP500', 'PHP1,000', 'PHP1,500', 'PHP2,000', 'PHP2,500', 'PHP3,000', 'PHP3,500', and 'PHP4,000'). The price range options were based on the approximate minimum-maximum price range of current vaccines in the Philippines.

Confidence in an international COVID-19 vaccine: Participants were asked to rate their level of confidence in using a vaccine for COVID-19 made in China, Russia, and the USA or Europe on a four-point scale ('completely not confident', 'not confident', 'confident', and 'completely confident'). Preference for the nationality of a manufacturer of the COVID-19 vaccine was also inquired.

## Ethics review and IRB approval

Our study protocol (Protocol Number 21-026) was reviewed and approved by the Institutional Review Board of Providence College on January 15, 2021. We had sought ethical review at the University of Santo Tomas in the Philippines but were advised by university authorities there to seek accelerated IRB approval in the United States because of the exigencies of the global pandemic. An informed consent statement was included in the survey instrument to welcome respondents who had clicked on the anonymous survey link provided by Qualtrics.

## Statistical analyses

All statistical analyses were conducted using Statistical Package for the Social Sciences (SPSS) version 27. A *p*-value of less than .05 was considered statistically significant. Frequency tables, charts, and proportions were used for data summarization—proportions and their respective 95% confidence intervals (CI) were calculated for each predictor variable. The model fit of binary logistic regression analysis was calculated using the Hosmer-Lemeshow

goodness-of-fit test (*Hosmer, Lemeshow & Sturdivant, 2013*). Participant responses to the one-item intent to receive COVID-19 vaccine ('If a vaccine for COVID-19 is available in the Philippines, would you use it?') was coded into two categories: vaccine hesitant (responses included: 'definitely no', 'probably no', 'unsure') and *not* vaccine hesitant (responses included: 'probably yes', 'definitely yes'). The eight options of WTP for a COVID-19 vaccine were categorized into three categories (PHP500–1,000, PHP1,500–2,500, PHP3,000–4,000). A multinomial logistic regression was employed to model factors associated with WTP for a COVID-19 vaccine with the lowest (PHP500–1,000) as the reference. We ran univariate analyses followed by a binary logistic regression analysis, including all factors showing significance ($p < .05$), to determine which factors predicted individual intention to receive a COVID-19 vaccine. Only significant factors in the univariate analyses were included in the binary logistic regression analysis.

## RESULTS

### Demographics

A total of 7,193 complete survey responses were received. Responses received represented participants with diverse demographics as shown in Table 1, and from all of the regions of the Philippines (Fig. 1). The study sample had a higher representation of younger adults aged 18 to 30 years old (52.4%), which is not unexpected given that our open access survey was deployed on social media. Additionally, the majority of participants identified as female (66.6%), single (65.7%), had obtained a college/university degree or above (84%), and lived in an urban location (78.9%). Only a small portion of the sample reported having an existing chronic condition (16.4%) while 17.5% reported having either 'very poor', 'poor', or 'fair' health. Additionally, 72.8% of the sample reported knowing someone who had tested positive for COVID-19.

### Health beliefs

As displayed in Fig. 2, with regards to perceived susceptibility to COVID-19 vaccines, the minority of participants reported that they thought that there was a high chance of personally contracting COVID-19 in the next few months (31.5%). However, when prompted to report worry about the likelihood of getting COVID-19, the majority of participants reported that they were worried (84.1%) and that COVID-19 is a serious illness with life-threatening conditions (96.3%). Furthermore, a significant majority (93.1%) reported that they were afraid of getting COVID-19, and that they would get very sick if they were infected with the virus (75%). The respondents in the survey reported significant perceived benefits for the COVID-19 vaccines. A large portion of participants noted that they believed that a COVID-19 vaccine would decrease the chances of getting COVID-19 (88.1%) and that the vaccine would alleviate their anxieties about catching the virus (84.5%). Notably, significant majorities of our respondents reported that they had worries about possible side-effects (89.6%), effectiveness (87.1%), safety (88.8%), and high cost (78%) of the vaccines. Nearly all were concerned about the possibility of fake jabs

**Table 1 Demographics and COVID-19 vaccine intent (N = 7,193).**

| | Overall N (%) | If a vaccine for COVID-19 is available in the Philippines, would you use it? | |
| --- | --- | --- | --- |
| | | Vaccine hesitant (Definitely no/Probably no/Unsure) $n = 2696$ (%) | Not vaccine hesitant (Probably yes/Definitely yes) $n = 4497$ (%) |
| *Demographics* | | | |
| Age group (years) | | | |
| 18–30 | 3,770 (52.4) | 1,405 (37.3) | 2,365 (62.7) |
| 31–40 | 815 (11.3) | 306 (37.5) | 509 (62.5) |
| 41–50 | 861 (12) | 335 (38.9) | 526 (61.1) |
| 51–60 | 900 (12.5) | 326 (36.2) | 574 (63.8) |
| 61–89 | 847 (11.8) | 324 (38.3) | 523 (61.7) |
| Gender | | | |
| Female | 4,789 (66.6) | 1,955 (40.8) | 2,834 (59.2) |
| Male | 2,404 (33.4) | 741 (30.8) | 1,663 (69.2) |
| Marital Status | | | |
| Single | 4,724 (65.7) | 1,719 (36.4) | 3,005 (63.6) |
| Married | 2,469 (34.3) | 977 (39.6) | 1,492 (60.4) |
| Highest education level | | | |
| Elementary school or below | 1 | 1 | |
| Junior high school | 46 (.6) | 29 (63) | 17 (37) |
| Senior high school | 1,102 (15.3) | 378 (34.3) | 724 (65.7) |
| College/university of above | 6,044 (84) | 2,288 (37.9) | 3,756 (62.1) |
| Occupation | | | |
| Blue collar worker | 91 (1.3) | 42 (46.2) | 49 (53.8) |
| Profession/white collar worker | 2,980 (41.4) | 1,133 (38) | 1,847 (62) |
| Self-employed | 575 (8) | 254 (44.2) | 321 (55.8) |
| Student | 2,831 (39.4) | 976 (34.5) | 1,855 (65.5) |
| Housewife/retired/unemployed/other | 716 (10) | 291 (40.6) | 425 (59.4) |
| Monthly income (PHP) | | | |
| ≤10,000 | 2,583 (35.9) | 1,004 (38.9) | 1,579 (61.1) |
| 10,000–20,000 | 954 (13.3) | 468 (49.1) | 486 (50.9) |
| 20,000–100,000 | 2,517 (35) | 907 (36) | 1,610 (64) |
| ≥100,000 | 1,139 (15.8) | 317 (27.8) | 822 (72.2) |
| Location | | | |
| Urban | 5,676 (78.9) | 1,985 (35) | 3,691 (65) |
| Rural | 1,517 (21.1) | 711 (46.9) | 806 (53.1) |
| Ever tested positive for COVID-19 | | | |
| Yes | 211 (2.9) | 68 (32.2) | 143 (67.8) |
| No | 6,982 (97.1) | 2,628 (37.6) | 4,354 (62.4) |

| | | If a vaccine for COVID-19 is available in the Philippines, would you use it? | |
| | Overall N (%) | Vaccine hesitant (Definitely no/Probably no/Unsure) *n* = 2696 (%) | Not vaccine hesitant (Probably yes/Definitely yes) *n* = 4497 (%) |
|---|---|---|---|
| Know anyone who has tested positive for COVID-19 | | | |
| Yes | 5,234 (72.8) | 1,780 (34) | 3,454 (66) |
| No | 1,959 (27.2) | 916 (46.8) | 1,043 (53.2) |
| Have an existing chronic condition | | | |
| Yes | 1,178 (16.4) | 396 (33.6) | 782 (66.4) |
| No | 6,015 (83.6) | 2,300 (38.2) | 3,715 (61.8) |
| Perceived overall health | | | |
| Very good | 1,997 (27.8) | 802 (40.2) | 1,195 (59.8) |
| Good | 3,934 (54.7) | 1,376 (35) | 2,558 (65) |
| Fair/Poor/Very poor | 1,262 (17.5) | 518 (41) | 744 (59) |

(97.4%). Many participants noted they would only receive the COVID-19 vaccines after many others had received it (72.8%) or after politicians had received it (68.2%).

## COVID-19 vaccination intent

Figure 3 shows the proportion of responses for intention to take a COVID-19 vaccine if one were available in the Philippines. A total of 4,497 of the participants (62.5%) responded either 'probably yes' or 'definitely yes' to COVID-19 vaccine intent—demonstrating that they were *not* vaccine hesitant—while 2,696 (37.4%) displayed vaccine hesitancy (responses included 'definitely no', 'probably no', and 'unsure'). More specifically, the majority of responses were 'probably yes' (32.8%, *n* = 2,358), followed by 'definitely yes' (29.7%, *n* = 2,139), 'unsure' (28%, *n* = 2,017), 'probably no' (6.4%, *n* = 461), and 'definitely no' (3%, *n* = 318). Demographics of respondents who intend (not vaccine hesitant) and do not intend (vaccine hesitant) to take a COVID-19 vaccine is displayed in Table 1.

Table 2 shows the univariate and binary analyses of factors associated with a vaccine hesitant and a not vaccine hesitant intention by demographics and health belief constructs. Univariate analyses showed a significantly higher proportion of participants who were single (63.6%) expressed an intention to take a COVID-19 vaccine (not vaccine hesitant) than married participants (60.4%). However, the association was not significant in the binary analysis. By occupational category, a significantly higher proportion of respondents that were not vaccine hesitant included those who identified as students (65.5%) and professional/white collar workers (62%). Significant differences were noted in vaccine hesitancy for COVID-19 by location, whereby individuals in an urban location (65%) reported a higher proportion of an intention to vaccinate compared to respondents in rural locations (53.1%).

By demographics, binary analyses revealed that males have greater odds of an intention to take a COVID-19 vaccine (OR = 1.222, 95% CI [1.078–1.386]) than females. Being

| REGION OF THE PHILIPPINES | Number of Responses (n) | Percent of Responses (%) |
| --- | --- | --- |
| BARMM | 15 | 0.2% |
| CAR | 41 | 0.6% |
| NCR | 3,602 | 50.1% |
| Region I | 131 | 1.8% |
| Region II | 125 | 1.7% |
| Region III | 722 | 10% |
| Region IV-A | 1,024 | 14.2% |
| Region IV-B | 56 | 0.8% |
| Region IX | 52 | 0.7% |
| Region V | 92 | 1.3% |
| Region VI | 456 | 6.3% |
| Region VII | 504 | 7% |
| Region VIII | 85 | 1.2% |
| Region X | 141 | 2% |
| Region XI | 79 | 1.1% |
| Region XII | 41 | 0.6% |
| Region XIII | 14 | 0.2% |
| No Region Listed | 13 | 0.2% |

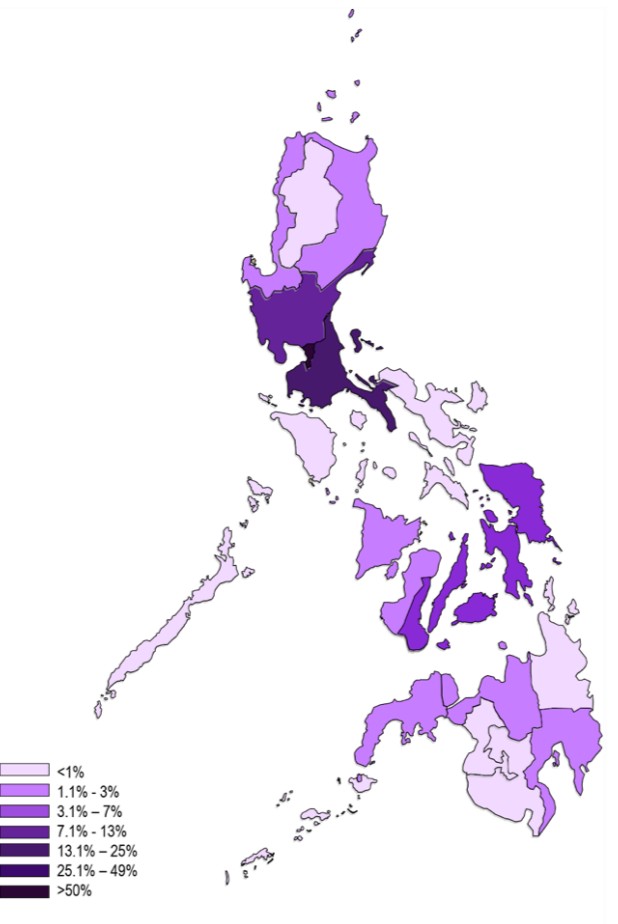

<1%
1.1% - 3%
3.1% – 7%
7.1% - 13%
13.1% – 25%
25.1% – 49%
>50%

**Figure 1** Geographical distribution of responses to our online survey.

self-employed (OR = .776, 95% CI [.586–1.026]), a student (OR = 1.352, 95% CI [1.046–1.749]), having a monthly income of less than PHP10,000 (OR = .596, 95% [CI, .477–.745]), PHP10,000–20,000 (OR = .587, 95% CI [.465–.741]), and PHP20,000–100,000 (OR = .822, 95% CI [.679–.995]) were also significant predictors of intent to vaccinate. Additionally, perceived overall health reported as 'fair', 'poor', and 'very poor' (OR = .755, 95% CI [.632–.903]), and 'good' (OR = 1.173, 95% CI [1.024, 1.343]) were significant predictors of intent to vaccinate for COVID-19.

Most of the constructs of the HBM were significantly associated with an intention to take a COVID-19 vaccination in the univariate analyses. While 72.8% of participants reported they would only take the COVID-19 vaccine after many others receive it (Fig. 3), disagreement with that notion (OR = 1.894, 95% CI [1.597–2.247]) was the strongest predictor for an intent to vaccinate. Intent to receive a COVID-19 vaccine only once politicians receive it (OR = 1.411, 95% CI [1.214–1.640]) was also a strong predictor of intention to take a COVID-19 vaccine.

**Table 2 Demographics, characteristics, and factors associated with an intention to take the COVID-19 vaccine (N=7,193).**

| | Overall N (%) | Univariable analysis | | | | Multivariable logistic regression | |
| --- | --- | --- | --- | --- | --- | --- | --- |
| | | Vaccine hesitant n = 2,696 | Not vaccine hesitant n = 4,497 | Unadjusted OR [95% CI] | p-value | Vaccine hesitant vs. Not vaccine hesitant Adjusted OR [95% CI] | p-value |
| *Demographics* | | | | | | | |
| Age group (years) | | | | | | | |
| 18–30 | 3,770 (52.4) | 1,405 (37.3) | 2,365 (62.7) | Reference | | | |
| 31–40 | 815 (11.3) | 306 (37.5) | 509 (62.5) | 1.043 [.894, 1.216] | .800 | | |
| 41–50 | 861 (12) | 335 (38.9) | 526 (61.1) | 1.030 [8.45 | | | |
| 51–60 | 900 (12.5) | 326 (36.2) | 574 (63.8) | .973 [.800, 1.182] | | | |
| 61–89 | 847 (11.8) | 324 (38.3) | 523 (61.7) | 1.091 [.898, 1.324] | | | |
| Gender | | | | | | | |
| Female | 4,789 (66.6) | 1,955 (40.8) | 2,834 (59.2) | .646 [.582, .717] | <.001 | Reference | |
| Male | 2,404 (33.4) | 741 (30.8) | 1,663 (69.2) | | | 1.223 [1.078, 1.386] | .002 |
| Marital status | | | | | | | |
| Single | 4,724 (65.7) | 1,719 (36.4) | 3,005 (63.6) | | | 1.098 [.939, 1.286] | .242 |
| Married | 2,469 (34.3) | 977 (39.6) | 1,492 (60.4) | .874 [.790, .966] | .008 | Reference | |
| Highest education level | | | | | | | |
| Elementary school or below | 1 | 1 | *n* too small to calculate | | | | |
| Junior high school | 46 (.6) | 29 (63) | 17 (37) | 2.80 [1.535, 5.107] | <.001 | .465 [.234, .924] | .029 |
| Senior high school | 1,102 (15.3) | 378 (34.3) | 724 (65.7) | .857 [.749, .981] | | 1.064 [.886, 1.277] | .506 |
| College/university of above | 6,044 (84) | 2,288 (37.9) | 3,756 (62.1) | Reference | | Reference | |
| Occupation | | | | | | | |
| Blue collar worker | 91 (1.3) | 42 (46.2) | 49 (53.8) | 1.252 [.808, 1.940] | <.001 | 1.057 [.624, 1.789] | .837 |
| Profession/white collar worker | 2,980 (41.4) | 1,133 (38) | 1,847 (62) | .896 [.759, 1.058] | | 1.051 [.845, 1.306] | .657 |
| Self-employed | 575 (8) | 254 (44.2) | 321 (55.8) | 1.156 [.925, 1.443] | | .776 [.586, 1.026] | .075 |
| Student | 2,831 (39.4) | 976 (34.5) | 1,855 (65.5) | .768 [.650, .909] | | 1.352 [1.046, 1.749] | .021 |
| Housewife/retired/unemployed/other | 716 (10) | 291 (40.6) | 425 (59.4) | Reference | | Reference | |
| Monthly income (PHP) | | | | | | | |
| ≤10,000 | 2,583 (35.9) | 1,004 (38.9) | 1,579 (61.1) | 1.649 [1.417, 1.919] | <.001 | .596 [.477, .745] | <.001 |
| 10,000–20,000 | 954 (13.3) | 468 (49.1) | 486 (50.9) | 2.497 [2.083, 2.994] | | .587 [.465, .741] | <.001 |
| 20,000–100,000 | 2,517 (35) | 907 (36) | 1,610 (64) | 1.461 [1.254, 1.702] | | .822 [.679, .995] | .044 |
| ≥100,000 | 1139 (15.8) | 317 (27.8) | 822 (72.2) | Reference | | Reference | |
| Location | | | | | | | |
| Urban | 5,676 (78.9) | 1,985 (35) | 3,691 (65) | | | Reference | |
| Rural | 1,517 (21.1) | 711 (46.9) | 806 (53.1) | 1.640 [1.463, 1.840] | <.001 | .728 [.634, .836] | <.001 |

Peer J

**Table 2** (*continued*)

| | Overall N (%) | Vaccine hesitant n = 2,696 | Not vaccine hesitant n = 4,497 | Unadjusted OR [95% CI] | p-value | Vaccine hesitant *vs.* Not vaccine hesitant Adjusted OR [95% CI] | p-value |
|---|---|---|---|---|---|---|---|
| | | **Univariable analysis** | | | | **Multivariable logistic regression** | |
| *Experience with COVID-19* | | | | | | | |
| Ever tested positive for COVID-19 | | | | | | | |
| Yes | 211 (2.9) | 68 (32.2) | 143 (67.8) | | | | |
| No | 6,982 (97.1) | 2,628 (37.6) | 4,354 (62.4) | .788 [.588, 1.056] | .110 | | |
| Know anyone who has tested positive for COVID-19 | | | | | | | |
| Yes | 5,234 (72.8) | 1,780 (34) | 3,454 (66) | | | 1.535 [1.348, 1.748] | <.001 |
| No | 1,959 (27.2) | 916 (46.8) | 1,043 (53.2) | .587 [.528, .652] | <.001 | Reference | |
| *Health Characteristics* | | | | | | | |
| Have an existing chronic condition | | | | | | | |
| Yes | 1,178 (16.4) | 396 (33.6) | 782 (66.4) | | | 1.149 [.964, 1.369] | .121 |
| No | 6,015 (83.6) | 2,300 (38.2) | 3,715 (61.8) | .818 [.717, .933] | .003 | Reference | |
| **Perceived overall health** | | | | | | | |
| Very good | 1,997 (27.8) | 802 (40.2) | 1,195 (59.8) | Reference | | Reference | |
| Good | 3,934 (54.7) | 1,376 (35) | 2,558 (65) | .802 [.717, .896] | <.001 | 1.173 [1.024, 1.343] | .021 |
| Fair/Poor/Very poor | 1,262 (17.5) | 518 (41) | 744 (59) | 1.037 [.899, 1.197] | | .755 [.632, .903] | .002 |
| *Health belief* | | | | | | | |
| **Perceived susceptibility** | | | | | | | |
| Chance of getting COVID-19 in the next few months is high | | | | | | | |
| Strongly agree/agree | 2,264 (31.5) | 1,595 (70.5) | 669 (29.5) | 1.665 [1.497, 1.853] | <.001 | Reference | |
| Disagree/strongly disagree | 4,929 (68.5) | 2,027 (41.1) | 2,902 (58.9) | | | .728 [.639, .830] | <.001 |
| Worry about the likelihood of getting COVID-19 | | | | | | | |
| Strongly agree/agree | 6,047 (84.1) | 2,105 (34.8) | 3,942 (65.2) | 1.994 [1.756, 2.265] | <.001 | Reference | |
| Disagree/strongly disagree | 1,146 (15.9) | 591 (51.6) | 555 (48.4) | | | .696 [.584, .830] | <.001 |

Caple et al. (2022), *PeerJ*, DOI 10.7717/peerj.12887

**Table 2** (*continued*)

| | Overall N (%) | Univariable analysis | | | p-value | Multivariable logistic regression | p-value |
|---|---|---|---|---|---|---|---|
| | | Vaccine hesitant n = 2,696 | Not vaccine hesitant n = 4,497 | Unadjusted OR [95% CI] | | Vaccine hesitant *vs.* Not vaccine hesitant Adjusted OR [95% CI] | |
| **Perceived severity** | | | | | | | |
| COVID-19 is serious with life-threatening complications | | | | | | | |
| Strongly agree/agree | 6,928 (96.3) | 2,523 (36.4) | 4,405 (63.6) | 3.283 [2.538, 4.248] | <.001 | Reference | |
| Disagree/strongly disagree | 265 (.04) | 173 (65.3) | 95 (34.7) | | | .580 [.410, .820] | .002 |
| I will be very sick if I get COVID-19 | | | | | | | |
| Strongly agree/agree | 5,394 (75) | 1,911 (35.4) | 3,483 (64.6) | 1.411 [1.266, 1.573] | <.001 | Reference | |
| Disagree/strongly disagree | 1,799 (25) | 785 (43.6) | 1,014 (56.4) | | | .824 [.711, .954] | .010 |
| I am afraid of getting COVID-19 | | | | | | | |
| Strongly agree/agree | 6,700 (93.1) | 2,439 (36.4) | 4,261 (63.6) | 1.902 [1.583, 2.286] | <.001 | Reference | |
| Disagree/strongly disagree | 493 (.07) | 257 (52.1) | 236 (47.9) | | | .676 [.513, .889] | .005 |
| **Perceived benefits** | | | | | | | |
| Vaccination will decrease my chances of getting COVID-19 | | | | | | | |
| Strongly agree/agree | 6,339 (88.1) | 2,006 (31.6) | 4,333 (68.4) | 9.088 [7.604, 10.862] | <.001 | Reference | |
| Disagree/strongly disagree | 854 (11.9) | 690 (80.8) | 164 (19.2) | | | .291 [.229, .370] | <.001 |
| Vaccination will decrease my worries about catching COVID-19 | | | | | | | |
| Strongly agree/agree | 6,078 (84.5) | 1,833 (30.2) | 4,245 (69.8) | 7.931 [6.822, 9.220] | <.001 | Reference | |
| Disagree/strongly disagree | 1,115 (15.5) | 863 (77.4) | 252 (22.6) | | | .265 [.218, .323] | <.001 |
| **Perceived barriers** | | | | | | | |
| Worry about the possible side-effects of the COVID-19 vaccine | | | | | | | |
| Strongly agree/agree | 6,447 (89.6) | 2,647 (41.1) | 3,800 (58.9) | .101 [.075, .135] | <.001 | Reference | |
| Disagree/strongly disagree | 746 (10.4) | 49 (6.6) | 697 (93.4) | | | 3.053 [2.107, 4.424] | <.001 |
| I worry about the effectiveness of the COVID-19 vaccine | | | | | | | |
| Strongly agree/agree | 6,263 (87.1) | 2,581 (41.2) | 3,682 (58.8) | .201 [.165, .246] | <.001 | Reference | |
| Disagree/strongly disagree | 930 (12.9) | 115 (12.4) | 815 (87.6) | | | 1.358 [1.022, 1.805] | .035 |

**Table 2** (*continued*)

| | Overall N (%) | Univariable analysis | | | | Multivariable logistic regression | |
| | | Vaccine hesitant *n* = 2, 696 | Not vaccine hesitant *n* = 4, 497 | Unadjusted OR [95% CI] | *p*-value | Vaccine hesitant *vs.* Not vaccine hesitant Adjusted OR [95% CI] | *p*-value |
| --- | --- | --- | --- | --- | --- | --- | --- |
| I worry about the safety of the COVID-19 vaccine | | | | | | | |
| Strongly agree/agree | 6,389 (88.8) | 2,626 (41.1) | 3,763 (58.9) | .137 [.106, .176] | <.001 | Reference | |
| Disagree/strongly disagree | 804 (11.2) | 70 (8.7) | 734 (91.3) | | | 1.593 [1.115, 2.277] | .011 |
| I worry about the high cost of the COVID-19 vaccine | | | | | | | |
| Strongly agree/agree | 5,608 (78) | 2,301 (41) | 3,307 (59) | .477 [.421, .541] | <.001 | Reference | |
| Disagree/strongly disagree | 1,585 (22) | 395 (24.9) | 1,190 (75.1) | | | 1.128 [.958, 1.327] | .147 |
| I worry about fake COVID-19 vaccines | | | | | | | |
| Strongly agree/agree | 7,006 (97.4) | 2,646 (37.8) | 4,360 (62.2) | .601 [.433, .834] | .002 | Reference | |
| Disagree/strongly disagree | 187 (.03) | 50 (26.7) | 197 (73.3) | | | .599 [.399, .900] | .014 |
| I worry that the COVID-19 vaccines will make me sick | | | | | | | |
| Strongly agree/agree | 4,918 (68.4) | 2,404 (48.9) | 2,514 (51.1) | .154 [.135, .176] | <.001 | Reference | |
| Disagree/strongly disagree | 2,275 (31.6) | 292 (12.8) | 1,983 (87.2) | | | 2.913 [2.467, 3.440] | <.001 |
| I worry the COVID-19 vaccines will not be effective against new virus variants | | | | | | | |
| Strongly agree/agree | 5,757 (80) | 2,506 (43.5) | 3,251 (56.5) | .198 [.168, .232] | <.001 | Reference | |
| Disagree/strongly disagree | 1,436 (20) | 190 (13.2) | 1,246 (86.8) | | | 1.831 [1.502, 2.232] | <.001 |
| **Cues to action** | | | | | | | |
| I will only receive the COVID-19 vaccines after many others receive it | | | | | | | |
| Strongly agree/agree | 5,237 (72.8) | 2,278 (43.5) | 2,959 (56.5) | .353 [.313, .398] | <.001 | Reference | |
| Disagree/strongly disagree | 1,956 (27.2) | 418 (21.4) | 1,538 (78.6) | | | 1.894 [1.597, 2.247] | <.001 |
| I will only receive the COVID-19 vaccines after politicians receive it | | | | | | | |
| Strongly agree/agree | 4,908 (68.2) | 2,124 (43.3) | 2,784 (56.7) | .438 [.392, .489] | <.001 | Reference | |
| Disagree/strongly disagree | 2,285 (31.8) | 572 (25) | 1,713 (75) | | | 1.411 [1.214, 1.640] | <.001 |

Hosmer–Lemeshow test, chi-square: 13.316, *p* = .101; Nagelkerke $R^2$ = .384.

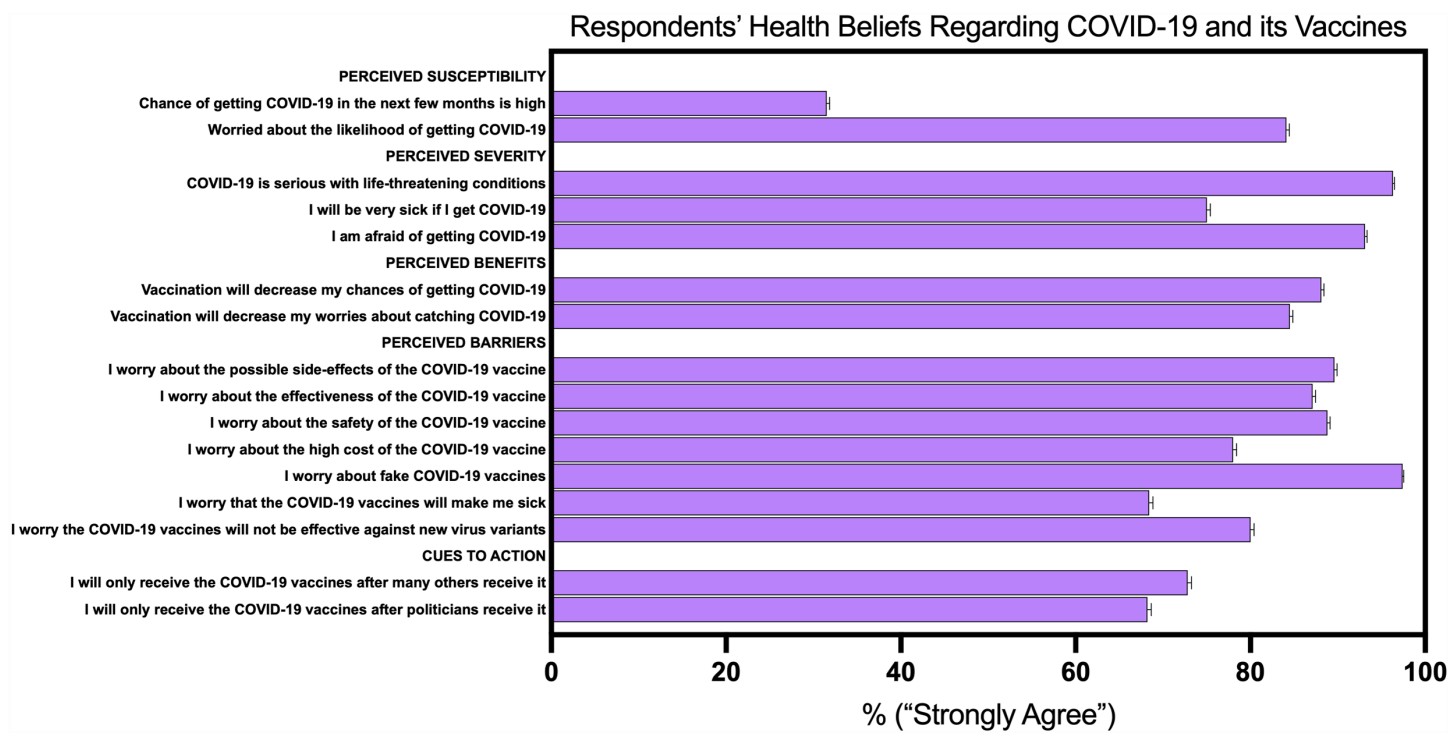

**Figure 2** Respondents' health beliefs regarding COVID-19 and its vaccines.

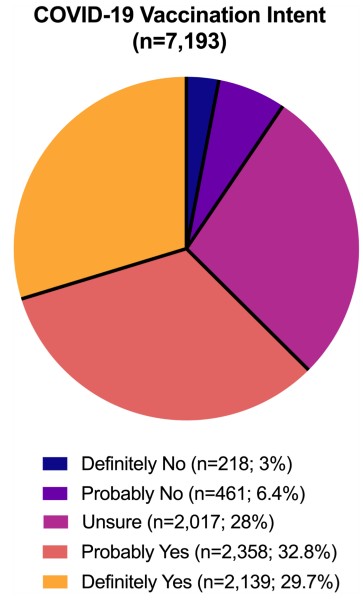

**COVID-19 Vaccination Intent**
**(n=7,193)**

■ Definitely No (n=218; 3%)
■ Probably No (n=461; 6.4%)
■ Unsure (n=2,017; 28%)
■ Probably Yes (n=2,358; 32.8%)
■ Definitely Yes (n=2,139; 29.7%)

**Figure 3** COVID-19 vaccination intent.

**Table 3  Willingness to pay for a COVID-19 vaccine.**

| Maximum amount willing to pay (PHP: Philippine Peso) | % of Respondents |
|---|---|
| PHP 500 | 18.8% |
| PHP 1,000 | 21% |
| PHP 1,500 | 12% |
| PHP 2,000 | 16% |
| PHP 2,500 | 6.8% |
| PHP 3,000 | 8.8% |
| PHP 3,500 | 1.9% |
| PHP 4,000 | 14.7% |

## Willingness to pay (WTP)

Table 3 shows that most participants were willing to pay PHP1,000 (USD20.38) (21%) followed by PHP500 (USD10.18) (18.8%) for two doses of a COVID-19 vaccine. The median (interquartile range [IQR]) of WTP for two doses of a COVID-19 vaccine was PHP2,000 (USD40.73). Table 4 shows the results of the univariate and multinomial regression analyses for the marginal WTP for an amount of PHP500/1,000 (USD10.18/20.36), PHP1,500/2,000/2,500 (USD30.55/40.73/50.91), PHP3000/3500/4000 (USD61.09/71.28/81.46) by demographics and HBM constructs. The results of the multinomial logistic regression (PHP1,500/2,000/2,500 *vs.* PHP500/1,000 and PHP3,000/3,500/4,000 *vs.* PHP500/1,000) revealed that individuals aged 31 to 40 displayed a higher WTP: PHP3,000/3,500/4,000 (USD61.09/71.28/81.46) over PHP500/1,000 (USD10.18/20.36). Compared to married participants, single respondents had the highest WTP: PHP3,000/3,500/4,000 (USD61.09/71.28/81.46) over PHP500/1,000 (USD10.18/20.36). Students had the highest WTP: PHP3,000/3,500/4,000 (USD61.09/71.28/81.46) over PHP500/1,000 (USD10.18/20.36). For monthly income, there was a gradual increase in the odds of WTP for an amount PHP1,500/2,000/2,500 (USD30.55/40.73/50.91) over PHP500/1,000 (USD10.18/20.36). Participants in rural locations were WTP: PHP1,500/2,000/2,500 (USD30.55/40.73/50.91) over PHP500/1,000 (USD10.18/20.36). For HBM constructs, similar to intent to vaccinate, a higher WTP was significantly associated with items in the perceived susceptibility to and severity of COVID-19, perceived benefits of a COVID-19 vaccine, perceived barriers, and cues to action constructs.

## Vaccine brand preference

Figure 4 shows confidence in foreign-made (*i.e.,* China, Russia, and the USA or Europe) COVID-19 vaccines. The vast majority of participants were 'confident' (59.7%) or 'completely confident' (23.1%) in a COVID-19 vaccine made in the USA or Europe. In contrast, a majority of participants indicated they were either 'completely not confident' (38.2%, 16.5%) or 'not confident' (46.8%, 49.2%) in a vaccine developed in China and Russia respectively. Findings on the preference of where a COVID-19 vaccine is made revealed respondents reported a preference for a vaccine made in the USA or Europe

**Table 4  Multinomial logistic regression of factors associated with marginal willingness-to-pay (WTP) for a COVID-19 vaccine (N = 7, 193).**

| | Univariable analysis | | | | Multinomial logistic regression | | | |
| | Marginal WTP (₱ = PHP) | | | | | | | |
| | ₱ 500/1000(US $10.18/20.36 )n = 2862 (%) | ₱ 1500/2000/2500 (US $30.55/40.73/50.91) n = 2505 (%) | ₱ 3000/3500/4000 (US $ 61.09/71.28/81.46) n = 1826 (%) | p-value | ₱1500/2000/2500 OR [95% CI] | p-value | ₱ 3000/3500/4000 OR [95% CI] | p-value |
|---|---|---|---|---|---|---|---|---|
| *Demographics* | | | | | | | | |
| Age group (years) | | | | | | | | |
| 18–30 | 1,349 (35.8) | 1,480 (39.3) | 941 (25) | <.001 | 1.895 [1.401, 2.563] | <.001 | 2.344 [1.616, 3.400] | <.001 |
| 31–40 | 352 (43.2) | 233 (28.6) | 230 (28.2) | | 1.960 [1.401, 2.611] | <.001 | 2.895 [2.045, 4.100] | <.001 |
| 41–50 | 363 (42.2) | 266 (30.9) | 232 (26.9) | | 1.640 [1.256, 2.143] | <.001 | 1.812 [1.309, 2.507] | <.001 |
| 51–60 | 418 (46.4) | 269 (29.9) | 213 (23.7) | | 1.261 [.982, 1.621] | .070 | 1.229 [.902, 1.675] | .192 |
| 61–89 | 380 (44.9) | 257 (30.3) | 210 (24.8) | | Reference | | Reference | |
| Gender | | | | | | | | |
| Female | 1,912 (39.9) | 1,682 (35.1) | 1,195 (25) | .391 | | | | |
| Male | 950 (39.5) | 823 (34.2) | 631 (26.2) | | | | | |
| Marital status | | | | | | | | |
| Single | 1,727 (36.6) | 1,768 (37.4) | 1,229 (26) | <.001 | 1.200 [1.003, 1.436] | .046 | 1.328 [1.069, 1.649] | .010 |
| Married | 1,135 (46) | 737 (29.9) | 597 (24.2) | | Reference | | Reference | |
| Highest education level | | | | | | | | |
| Elementary school or below | 1 | | | | *n* too small to calculate | | | |
| Junior high school | 19 (41.3) | 14 (30.4) | 13 (28.3) | | .838 [.396, 1.771] | .643 | 1.958 [.899, 4.264] | .091 |
| Senior high school | 328 (29.8) | 474 (43) | 300 (27.2) | <.001 | 1.153 [.956, 1.390] | .136 | 1.203 [.958, 1.510] | .111 |
| College/university of above | 2,514 (41.6) | 2,017 (33.4) | 1,513 (25) | | Reference | | Reference | |
| Occupation | | | | | | | | |
| Blue collar worker | 69 (75.8) | 13 (14.3) | 9 (9.9) | <.001 | .238 [.124, .456] | <.001 | .304 [.135, .681] | .004 |
| Professional/white collar worker | 1,382 (46.4) | 845 (28.4) | 753 (25.3) | | .681 [.539, .860] | .001 | .866 [.644, 1.163] | .339 |
| Self-employed | 226 (39.3) | 204 (35.5) | 145 (25.2) | | 1.052 [.795, 1.393] | .722 | 1.127 [.785, 1.616] | .517 |
| Student | 585 (30.3) | 1213 (42.8) | 760 (26.8) | | 1.614 [1.206, 2.159] | .001 | 1.725 [1.190, 2.501] | .004 |
| Housewife/retired/unemployed/other | 327 (45.7) | 230 (32.1) | 159 (22.2) | | Reference | | Reference | |
| Monthly income (PHP) | | | | | | | | |
| ≤10,000 | 986 (38.2) | 995 (38.5) | 602 (23.3) | | .378 [.298, .480] | <.001 | .252 [.191, .333] | <.001 |
| 10,000–20,000 | 554 (58.1) | 272 (28.5) | 128 (13.4) | <.001 | .389 [.305, .497] | <.001 | .211 [.156, .286] | <.001 |
| 20,000–100,000 | 1047 (41.6) | 849 (33.7) | 621 (24.7) | | .614 [.503, .749] | <.001 | .452 [.362, .563] | <.001 |
| ≥100,000 | 275 (24.1) | 389 (34.2) | 475 (41.7) | | Reference | | Reference | |
| Location | | | | | | | | |
| Urban | 2,053 (36.2) | 2,068 (36.4) | 1,555 (27.4) | | Reference | | Reference | |
| Rural | 809 (53.3) | 437 (28.8) | 271 (17.9) | <.001 | .584 [.505, .674] | <.001 | .552 [.460, .662] | <.001 |

**Table 4** (*continued*)

| | Univariable analysis | | | | Multinomial logistic regression | | | |
|---|---|---|---|---|---|---|---|---|
| | Marginal WTP (₱ = PHP) | | | | | | | |
| | ₱ 500/1000(US $10.18/ 20.36 )n = 2862 (%) | ₱ 1500/2000 /2500 (US $30.55/40.73 /50.91) n = 2505 (%) | ₱ 3000/3500 /4000 (US $ 61.09/71.28 /81.46) n = 1826 (%) | *p*-value | ₱ 1500/2000 /2500 OR [95% CI] | *p*-value | ₱ 3000/3500 /4000 OR [95% CI] | *p*-value |
| *Experience with COVID-19* | | | | | | | | |
| Ever tested positive for COVID-19 | | | | | | | | |
| Yes | 80 (37.9) | 66 (31.3) | 65 (30.8) | .176 | | | | |
| No | 2,782 (39.8) | 2,439 (34.9) | 1,761 (25.2) | | | | | |
| Know anyone who has tested positive for COVID-19 | | | | | | | | |
| Yes | 1,932 (36.9) | 1,868 (35.7) | 1,434 (27.4) | <.001 | 1.336 [1.169, 1.527] | <.001 | 1.419 [1.201, 1.678] | <.001 |
| No | 930 (47.5) | 637 (32.5) | 392 (20) | | Reference | | Reference | |
| *Health Characteristics* | | | | | | | | |
| Have an existing chronic condition | | | | | | | | |
| Yes | 438 (37.2) | 387 (32.9) | 353 (30) | .001 | 1.176 [.981, 1.410] | .080 | 1.430 [1.152, 1.775] | .001 |
| No | 2,424 (40.3) | 2,118 (35.2) | 1,473 (24.5) | | Reference | | Reference | |
| **Perceived overall health** | | | | | | | | |
| Very good | 852 (42.7) | 641 (32.1) | 504 (25.2) | | Reference | | Reference | |
| Good | 1,559 (39.6) | 1,385 (35.2) | 990 (25.2) | | 1.160 [1.010, 1.332] | .035 | 1.052 [.891, 1.243] | .549 |
| Fair/Poor/Very poor | 451 (35.7) | 479 (38) | 332 (26.3) | .020 | 1.287 [1.071, 1.547] | .007 | 1.090 [.872, 1.364] | .450 |
| *Health belief* | | | | | | | | |
| **Perceived susceptibility** | | | | | | | | |
| Chance of getting COVID-19 in the next few months is high | | | | | | | | |
| Strongly agree/agree | 838 (37) | 790 (34.9) | 636 (28.1) | <.001 | Reference | | Reference | |
| Disagree/strongly disagree | 2,024 (41.1) | 1,715 (34.8) | 1,190 (24.1) | | .989 [.868, 1.128] | .872 | .916 [.783, 1.071] | .270 |
| Worry about the likelihood of getting COVID-19 | | | | | | | | |
| Strongly agree/agree | 2,272 (37.6) | 2,175 (36) | 1,600 (26.5) | <.001 | Reference | | Reference | |
| Disagree/strongly disagree | 590 (51.5) | 330 (28.8) | 226 (19.7) | | .763 [.638, .912] | .003 | .690 [.551, .866] | .001 |
| **Perceived severity** | | | | | | | | |
| COVID-19 is serious with life-threatening complications | | | | | | | | |
| Strongly agree/agree | 2,699 (39) | 2,441 (35.2) | 1,788 (25.8) | <.001 | Reference | | Reference | |
| Disagree/strongly disagree | 163 (61.5) | 64 (24.2) | 38 (14.3) | | .681 [.484, .960] | .028 | .562 [.654, .891] | .014 |

**Table 4** (*continued*)

| | Univariable analysis | | | | Multinomial logistic regression | | | |
|---|---|---|---|---|---|---|---|---|
| | Marginal WTP (₱ = PHP) | | | | | | | |
| | ₱ 500/1000(US $10.18/ 20.36 )n = 2862 (%) | ₱ 1500/2000 /2500 (US $30.55/40.73 /50.91) *n* = 2505 (%) | ₱ 3000/3500 /4000 (US $ 61.09/71.28 /81.46) *n* = 1826 (%) | *p*-value | ₱ 1500/2000 /2500 OR [95% CI] | *p*-value | ₱ 3000/3500 /4000 OR [95% CI] | *p*-value |
| I will be very sick if I get COVID-19 | | | | | | | | |
| Strongly agree/agree | 2,048 (38) | 1,931 (35.8) | 1,415 (26.2) | <.001 | Reference | | Reference | |
| Disagree/strongly disagree | 814 (45.2) | 574 (31.9) | 411 (22.8) | | .883 [.762, 1.023] | .097 | .784 [.654, .940] | .009 |
| I am afraid of getting COVID-19 | | | | | | | | |
| Strongly agree/agree | 2,619 (39.1) | 2,354 (35.1) | 1,727 (25.8) | <.001 | Reference | | Reference | |
| Disagree/strongly disagree | 243 (49.3) | 151 (30.6) | 99 (20.1) | | .989 [.759, 1.290] | .938 | .794 [.562, 1.122] | .192 |
| **Perceived benefits** | | | | | | | | |
| Vaccination will decrease my chances of getting COVID-19 | | | | | | | | |
| Strongly agree/agree | 2,341 (36.9) | 2,289 (36.1) | 1,709 (27) | <.001 | Reference | | Reference | |
| Disagree/strongly disagree | 521 (61) | 216 (25.3) | 117 (13.7) | | .666 [.526, .842] | .001 | .532 [.391, .724] | <.001 |
| Vaccination will decrease my worries about catching COVID-19 | | | | | | | | |
| Strongly agree/agree | 2,248 (37) | 2,180 (35.9) | 1,650 (27.1) | <.001 | Reference | | Reference | |
| Disagree/strongly disagree | 614 (55.1) | 325 (29.1) | 176 (15.8) | | | | .688 [.529, .895] | .005 |
| **Perceived barriers** | | | | | | | | |
| Worry about the possible side-effects of the COVID-19 vaccine | | | | | | | | |
| Strongly agree/agree | 2,701 (41.9) | 2,212 (34.3) | 1,534 (23.8) | <.001 | Reference | | Reference | |
| Disagree/strongly disagree | 161 (21.6) | 293 (39.3) | 292 (39.1) | | 1.421 [1.103, 1.831] | .007 | 1.262 [.945, 1.684] | .114 |
| I worry about the effectiveness of the COVID-19 vaccine | | | | | | | | |
| Strongly agree/agree | 2,590 (41.4) | 2,184 (34.9) | 1,489 (23.8) | <.001 | Reference | | Reference | |
| Disagree/strongly disagree | 272 (29.2) | 321 (34.5) | 337 (36.2) | | .917 [.712, 1.182] | .505 | .756 [.564, 1.015] | .063 |
| I worry about the safety of the COVID-19 vaccine | | | | | | | | |
| Strongly agree/agree | 2,637 (41.3) | 2,240 (35.1) | 1,512 (23.7) | <.001 | Reference | | Reference | |
| Disagree/strongly disagree | 225 (28) | 265 (33) | 314 (39.1) | | .768 [.578, 1.022] | .070 | .895 [.645, 1.243] | .508 |
| I worry about the high cost of the COVID-19 vaccine | | | | | | | | |
| Strongly agree/agree | 2,616 (46.6) | 1,991 (35.5) | 1,001 (17.8) | <.001 | Reference | | Reference | |
| Disagree/strongly disagree | 246 (15.5) | 514 (32.4) | 825 (52.1) | | 2.534 [2.107, 3.048] | <.001 | 7.321 [6.065, 8.836] | <.001 |

**Table 4** (*continued*)

| | Univariable analysis | | | | Multinomial logistic regression | | | |
|---|---|---|---|---|---|---|---|---|
| | Marginal WTP (₱ = PHP) | | | | | | | |
| | ₱ 500/1000 (US $10.18/20.36 ) n = 2862 (%) | ₱ 1500/2000/2500 (US $30.55/40.73/50.91) n = 2505 (%) | ₱ 3000/3500/4000 (US $ 61.09/71.28/81.46) n = 1826 (%) | *p*-value | ₱ 1500/2000/2500 OR [95% CI] | *p*-value | ₱ 3000/3500/4000 OR [95% CI] | *p*-value |
| I worry about fake COVID-19 vaccines | | | | | | | | |
| Strongly agree/agree | 2,809 (40.1) | 2,452 (35) | 1,745 (24.9) | <.001 | Reference | | Reference | |
| Disagree/strongly disagree | 53 (28.3) | 53 (28.3) | 81 (43.3) | | .723 [.471, 1.108] | .136 | 1.093 [.707, 1.690] | .688 |
| I worry that the COVID-19 vaccines will make me sick | | | | | | | | |
| Strongly agree/agree | 2,254 (45.8) | 1,640 (33.3) | 1,024 (20.8) | <.001 | Reference | | Reference | |
| Disagree/strongly disagree | 608 (26.7) | 865 (38) | 802 (35.3) | | 1.407 [1.200, 1.650] | <.001 | 1.431 [1.188, 1.725] | <.001 |
| I worry the COVID-19 vaccines will not be effective against new virus variants | | | | | | | | |
| Strongly agree/agree | 2,463 (42.8) | 1,984 (34.5) | 1,310 (22.8) | <.001 | Reference | | Reference | |
| Disagree/strongly disagree | 399 (27.8) | 521 (36.3) | 516 (35.9) | | 1.109 [.927, 1.327] | .259 | 1.245 [1.016, 1.525] | .035 |
| **Cues to action** | | | | | | | | |
| I will only receive the COVID-19 vaccines after many others receive it | | | | | | | | |
| Strongly agree/agree | 2,235 (42.7) | 1,833 (35) | 1,169 (22.3) | <.001 | Reference | | Reference | |
| Disagree/strongly disagree | 672 (34.4) | 657 (33.6) | | | 1.068 [.909, 1.254] | .425 | 1.103 [.913, 1.333] | .309 |
| I will only receive the COVID-19 vaccines after politicians receive it | | | | | | | | |
| Strongly agree/agree | 2,142 (43.6) | 1,725 (35.1) | 1,041 (21.2) | <.001 | Reference | | Reference | |
| Disagree/strongly disagree | 720 (31.5) | 780 (34.1) | 785 (34.4) | | 1.111 [.957, 1.289] | .167 | 1.495 [1.256, 1.780] | <.001 |

Goodness-of-Fit test, Pearson chi-square: 11670.557, *p* = .036; Nagelkerke $R^2$ = .264
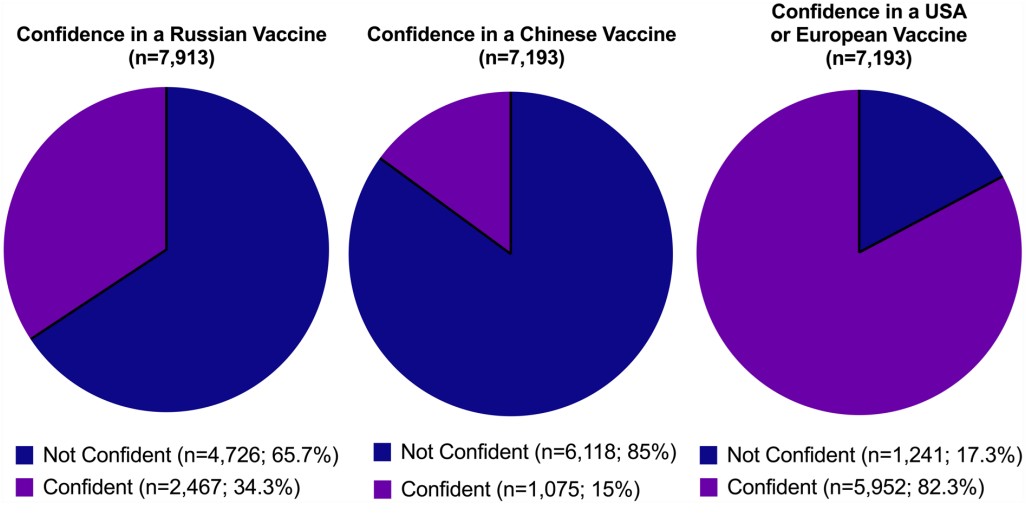

**Figure 4 Confidence in foreign-made COVID-19 vaccines.**

(53.4%) while 44.6% of participants indicated no preference of where a COVID-19 vaccine is made as long as it is safe and effective.

## DISCUSSION

In this study, we sought to interrogate the extent of COVID-19 vaccine hesitancy in the Philippines and to determine whether the Health Belief Model (HBM) could be used to explain this hesitancy among Filipinos. We deployed our nationwide open-access online survey for a two week period (January 15–29, 2021), a month before the first COVID-19 vaccines were administered in the archipelago on March 1, 2021.

We received nearly 7,200 completed surveys from around the country. The majority of responses (5,348; 74%) were from the three geographical and administrative regions, National Capital Region (NCR), Region III, and Region IVA, that encompass and surround the capital city of Manila. Together, these three regions, which have 38% of the population, have weathered the worst of the COVID-19 pandemic in the Philippines with about 60% of the total nationwide reported cases (https://doh.gov.ph/covid19tracker).

A majority of our respondents (62.5%) indicated that they were willing to be vaccinated by responding either 'probably yes' or 'definitely yes' to COVID-19 vaccine intent. As a point of comparison, a survey conducted by Pulse Asia from February 22, 2021, to March 3, 2021, which was a month after our survey period, reported that only 16% of the 2,400 Filipinos they interviewed face-to-face indicated that they would have themselves vaccinated, while 23% said that they "cannot say" if they would have themselves vaccinated (*Pulse Asia, 2021*).

There are many possible reasons for this difference in reported vaccine confidence but three immediately come to mind. First, our survey was an open access online survey while the Pulse Asia survey involved face-to-face interviews of Filipinos throughout the country.

By their very nature, online surveys are not representative of the population at large because access to the internet is uneven, especially in developing countries. Second, the respondents of our survey were skewed in favor of individuals living in those urban centers around the capital that have endured the most stringent quarantine restrictions of the pandemic. It is likely that this pandemic experience would have heightened their desire to be vaccinated as compared to those Filipinos who live in the countryside where viral transmission was sporadic and pandemic restrictions were relatively innocuous. Notably, our bivariate analysis confirms that individuals in an urban location (65%) reported a higher proportion of an intention to vaccinate compared to respondents in rural locations (53.1%). Finally, the intervening month between the two surveys witnessed several political events prior to the vaccine rollout that could have changed the public's views on the COVID-19 vaccines. Most significantly, on February 26, 2021, President Rodrigo R. Duterte signed into law a bill giving indemnity to vaccine makers should their vaccines cause serious adverse side effects among recipients. The bill was criticized by several senators of the Philippines who described it as a desperate move on the part of the Duterte administration to purchase untested vaccines on the global market. This political squabble could have decreased public confidence in the safety of the COVID-19 vaccines.

Multivariable analysis revealed that HBM constructs were associated with vaccination intention in the Philippines, which is in accordance with other studies from the Asia-Pacific region (*Wong et al., 2020*; *Wong et al., 2021*; *Lin et al., 2020*; *Yu et al., 2021*; *Kabir et al., 2021*; *Huynh et al., 2021*; *Tao, Wang & Liu, 2021*) Perceptions of high susceptibility, high severity, and significant benefits were all good predictors for vaccination intent. A study to interrogate vaccine hesitancy among Filipinos in two urban communities in Manila before the COVID-19 pandemic also found that respondents who believed in the protective nature of vaccines were less likely to report vaccine hesitancy and were nine times less likely to refuse vaccination for their children because of negative media exposure (*Migriño et al., 2020*).

Perceived barriers against COVID-19 immunization reported by our respondents including worries about the side-effects, effectiveness, and safety of the vaccines have also been reported by these other HBM studies. These are triggers for hesitancy that can be eradicated with scientific explanation. Public health authorities in the Philippines should address these issues. In response to the findings of this survey, we have initiated a public awareness campaign in the Philippines to directly respond to these concerns by generating infographics and other publication materials to alleviate these worries (https://www.facebook.com/USTCoVAX/).

Interestingly, we discovered that our Filipino respondents were overwhelmingly worried about fake COVID-19 vaccines (97.4%). This is not surprising given the prevalence of counterfeit items in Philippine society (*Calunsod, 2013*). Similar concerns have been raised in India (*Choudhary et al., 2021*) though this was not observed in China (*Lin et al., 2020*). This suggests that the national governments of developing countries should ensure the integrity of the vaccine rollout to reassure their citizens that they are not receiving "dud" doses.

Next, we found that external cues to action were important. Large majorities of our respondents would only receive the COVID-19 vaccines after many others had received it (72.8%) or after politicians had received it (68.2%). We observed that disagreement with the statement that the individual would receive the COVID-19 vaccine only after many others had received it was the strongest predictor for an intent to vaccinate among our Filipino respondents. This segment of the population could represent citizens who so want to be vaccinated that they are willing to put aside the collectivist mindset that is strongly rooted in Filipino culture (*Grimm et al., 1999*). However, given the high numbers of respondents who indicated that they were waiting for others to first receive the vaccine, our UST-CoVAX public awareness program began sharing the personal vaccination testimonies of Filipinos around the world on social media platforms to show Filipinos in the Philippines that others like them had already received the COVID-19 vaccines (https://www.facebook.com/USTCoVAX/).

Our study revealed that most (21%) were willing to pay an amount of PHP1,000 [USD20] for two doses of the COVID-19 vaccines with an average willing-to-pay amount of PHP1,892 (USD38). Multinomial logistic regression showed that individuals aged 31 to 40, single respondents, and students had the highest WTP in their demographic categories respectively. Since the minimum daily wage in the Philippines in 2021 is PHP537 [USD10.54], the average WTP amount of PHP1,892 (USD38) remains a significant investment in the health of the individual, equivalent to nearly four days of wages. This suggests that the COVID-19 vaccines should be provided free of charge to ensure population-wide access among all Filipinos across the economic classes.

Finally, our analysis revealed significant vaccine brand preference among our Filipino respondents. The vast majority of participants were 'confident' (59.7%) or 'completely confident' (23.1%) in a COVID-19 vaccine made in the USA or Europe. In contrast, a majority of participants indicated they were either 'completely not confident' (38.2%, 16.5%) or 'not confident' (46.8%, 49.2%) in a vaccine developed in China and Russia respectively. These findings mirror those reported by the Pulse Asia survey already described above that showed that a majority (52%) of Filipinos who were opting to get vaccinated preferred the Pfizer vaccine (*Pulse Asia, 2021*). The roots of this brand preference are not clear. One possibility could be the political controversy in the Philippine Senate where senators deemed the Chinese vaccine brands "unacceptable" because of their low efficacy (*Romero, 2020*). Anecdotally, Filipino social media influencers have also reminded Filipinos of the contaminated Chinese milk products that had been banned in the Philippines over a decade ago (*Crisostomo, 2012*). Regardless of the reasons, this vaccine preference has to be managed by the national government to prevent Filipinos from unnecessarily delaying immunization to obtain their preferred vaccine brand.

Our study has several limitations. As we already noted above, the use of an open-access online survey may result in sampling bias so we cannot generalize our findings to the entire Filipino population (*Wyatt, 2000*; *Eysenbach & Wyatt, 2002*). It is notable that young people aged 18–30 years, who make up around 28% of the population of the Philippines (https://www.populationpyramid.net/philippines/), constitute 52.4% of our respondents. Unexpectedly, however, senior citizens aged 61–89 years of age, who

constitute 8% of the country's population are also over-represented with 11.8% of the respondents. Furthermore, the responses were based on self-report and may be subject to self-reporting bias and a tendency to report socially desirable responses especially in a strongly collectivist society like the Philippines. One final limitation of our study is the bias associated with the assessment of acceptance and WTP for a hypothetical COVID-19 vaccine before any concrete vaccines actually exist (*Schmidt & Bijmolt, 2019*). We therefore intend to undertake a follow-up survey once the vaccine rollout in the country has stabilized. Nonetheless, despite these shortcomings, we believe that our findings will provide insights to support the vaccine rollout of the COVID-19 vaccines in the Philippines by helping public health authorities to understand vaccine demand and vaccine hesitancy in the country. Indeed, based on these findings, we inaugurated the UST-CoVAX public awareness campaign that seeks to increase vaccine confidence in the Philippines by addressing the specific fears and concerns of our Filipino respondents and by sharing the personal vaccination testimonies of Filipinos around the world (https://www.facebook.com/USTCoVAX/).

### Funding
The authors received no funding for this work.

### Competing Interests
The authors declare there are no competing interests.

### Author Contributions
- Alexandria Caple analyzed the data, prepared figures and/or tables, authored or reviewed drafts of the paper, and approved the final draft.
- Arnie Dimaano, Marc Martin Sagolili and April Anne Uy conceived and designed the experiments, analyzed the data, authored or reviewed drafts of the paper, and approved the final draft.
- Panjee Mariel Aguirre, Dean Lotus Alano, Giselle Sophia Camaya, Brent John Ciriaco, Princess Jerah Mae Clavo, Dominic Cuyugan, Paul Jeremy Lanete, Ardwayne Jurel La Torre, Thomas Loteyro, Raisa Mikaela Lua, Nicole Gayle Manansala, Raphael Willard Mosquito, Alexandra Erika Orfanel, Gheyanna Merly Pascual, Aubrey Joy Sale, Sophia Lorraine Tendenilla, Maria Sofia Lauren Trinidad, Nicole Jan Trinidad and Daphne Louise Verano conceived and designed the experiments, authored or reviewed drafts of the paper, and approved the final draft.
- Cleinne Florence Geeseler Fermo conceived and designed the experiments, prepared figures and/or tables, authored or reviewed drafts of the paper, and approved the final draft.
- Nicanor Austriaco conceived and designed the experiments, analyzed the data, prepared figures and/or tables, authored or reviewed drafts of the paper, and approved the final draft.

## Human Ethics

The following information was supplied relating to ethical approvals (i.e., approving body and any reference numbers):

Our study protocol (Protocol Number 21-026) was reviewed and approved by the Institutional Review Board of Providence College on January 15, 2021.

## Data Availability

The raw data from the 7,193 survey responses are available in the Supplemental Files.

## Supplemental Information

Supplemental information for this article can be found online at http://dx.doi.org/10.7717/peerj.12887#supplemental-information.

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
