# Peer review of "Interrogating COVID-19 vaccine intent in the Philippines with a nationwide open-access online survey"

_PeerJ, doi:10.7717/peerj.12887_

## Round 0.1 · original submission · Minor Revisions

Please follow Reviewer's suggestions for minor revisions.

·

Basic reporting

see attached PDF file - there are comments to help make the article clearer.

Experimental design

no comment

Validity of the findings

no comment

Reviewer 2 ·

Basic reporting

Generally, it is a well-written paper on an important issue, using professional English. The introduction part clearly highlights the meaning of the current work by introducing and summarizing relevant literature in a concise manner, the method used has been explained with details, and findings have been well discussed, explained, and compared with relevant studies in the discussion part.
One suggestion: content in lines 86-97 should not be put in the last paragraph of the introduction.
My main concern is that the online survey asked people's vaccination intention, cue to action about COVID-19 vaccination, and measured HBM constructs. However, the study is titled with vaccine hesitancy, and the main aim has been set around the core term — vaccine hesitancy. However, though the definition of VH has been developed quite broad, I don’t agree with the use of the term in the title and as the main aim in the current online survey, as you didn’t strictly design your questionnaire according to the definition of VH, like the “3C” definition of VH or “5C” definition of VH. It is a study that used HBM as a framework to link some important relevant psychological constructs to people’s vaccination intention and willingness to pay.

Experimental design

It is a well-designed online survey, the sample size is large, particularly the author showed a map of respondents' geographical distribution in the country, which is very convincing for the readers to believe the scale and possible representativeness of the survey. Limitations regarding the obvious systematic bias caused by using an online survey need well explanation — inevitably you exclude all that do not use the internet or do not want to participate in an online survey.

Validity of the findings

The study is a good addition to the international literature on this topic. The findings based on the online survey are clear and the potential bias has been well discussed.

---

## Round 0.2 · Major Revisions

Please remove Vergara as a cited reference recently added (post first review) from the manuscript and answer to Reviewer's comment.

·

Basic reporting

No Comment

Experimental design

No Comment

Validity of the findings

No comment

Additional comments

The use of Vergara as a cited reference recently added by the author post first review of manuscript is being raised as to the accuracy of the information provided by Vergara. It is because of this that I thought the authors might reflect on accuracy of cited reference carefully first before using it.

Dengvaxia in the Philippines. The use was for aged 9 year old and above. Vergara alluded that 600 infant deaths were due to this vaccine. The conclusion of Vergara as to no Industry or government officials punished seemed to reflect a biased judgement by Vergara when there is no judgement made in courts.

---

## Round 0.3 · accepted · Accept

The revision has been properly performed.

·

Basic reporting

no comment

Experimental design

no comment

Validity of the findings

No comment

Additional comments

Great paper that can help inform national policy and decision makers involved with management of COVID Pandemic. Likewise it can also be of help to those involved with health promotions, access to vaccines (e.g. industry).